# Molecular Insight into Genetic Structure and Diversity of Putative Hybrid Swarms of *Pinus sylvestris* × *P. mugo* in Slovakia

Miroslav Klobucnik [1,2,†], Martin Galgoci [1,3,†], Dusan Gomory [4] and Andrej Kormutak [1,*]

[1] Institute of Plant Genetics and Biotechnology, Plant Science and Biodiversity Center SAS, Akademicka 2, P.O. Box 39A, SK-950 07 Nitra, Slovakia; miroslav.klobucnik@savba.sk (M.K.); martin.galgoci@savba.sk (M.G.)

[2] Faculty of Natural Sciences, Comenius University, Ilkovicova 6, SK-842 15 Bratislava, Slovakia

[3] Faculty of Natural Sciences, Constantine the Philosopher University in Nitra, Trieda Andreja Hlinku 1, SK-949 74 Nitra, Slovakia

[4] Faculty of Forestry, Technical University in Zvolen, T.G. Masaryka 24, SK-960 53 Zvolen, Slovakia; dusan.gomory@tuzvo.sk

\* Correspondence: nrgrkorm@savba.sk

† These authors contributed equally to this work.

**Abstract:** The genetic structures of the four putative hybrid swarms of *Pinus sylvestris* × *P. mugo* in Slovakia were analyzed in terms of individual admixture proportions calculated via inter-primer binding site (iPBS) marker loci. This work aimed to reevaluate the hybrid swarms' differentiation status as postulated in the previous studies at both population and genomic levels. The study confirmed intermediate divergence of each of the swarms examined. Based on 80-loci hybrid index scores, we have revealed the presence of introgressive and intermediate hybrids with frequencies corresponding to differentiation estimates. Surprisingly, irrespective of individual phenotypes, the most frequent intermediates were found in Sucha Hora (29.5%) and Obsivanka (28.6%) populations, which resemble rather pure *P. mugo* and were previously considered as bimodal hybrid zones with a negligible amount of hybrids. The remaining hybrid zone population Zuberec seems to be highly introgressed to *P. sylvestris*, while Tisovnica is clearly inclined to *P. mugo*. The results and different methodologies are discussed.

**Keywords:** *Pinus sylvestris*; *P. mugo*; hybrids; genetic structure; hybrid index

## 1. Introduction

Studies addressing natural hybrid populations belong among the most important discoveries of the early 20th century, as they have laid the foundation of the modern evolutionary concept of hybridization, along with allopolyploidy and recombination and speciation [1]. With special reference to the genus Pinus, hybridization and introgression are important attributes of the reproductive behavior of pines and an indisputable part of their evolutionary history [2]. Hybridization may contribute to speciation through the formation of new hybrid taxa, whereas the introgression within a few loci may promote the adaptive divergence and facilitate speciation [3]. Among seven introgressive hybrids of pines described within a group of hard pines and two introgressive hybrids revealed in a group of soft pines, the hybrid swarms of Scots pine (*P. sylvestris* L.) and dwarf mountain pine (*P. mugo* Turra s. str.) dominate ecologically in Central and Southern Europe [4,5]. As typical pioneer woody plants, their parental species occupy sunny to partially shaded, nutrient-poor sites from lowlands and foothills (*P. sylvestris*) to the subalpine belt of the Eastern Alps and the Carpathians (*P. mugo* s. str.). However, being very undemanding as to edaphic conditions and water supply, they may also withstand extreme habitats, such as anoxic peatlands in lower elevations, due to their adaptation to low nutrients and light

availability [6]. Indeed, it is this habitat that is considered the most common site that favors extrazonal occurrence of *P. mugo* and spontaneous hybridization with *P. sylvestris* [7].

Systematically, the species are considered closely related representatives of the subgenus Pinus, subsection Sylvestres [4]. Hybridization between these species was postulated to be one of the most significant evolutionary processes leading to formation of the new taxa within the *Pinus mugo* complex [8]. The latter involves *P. mugo* Turra (s. str.) and *P. uncinata* Ram. ex DC. subsp. *uncinata*, as well as one hybrid taxon, *P. mugo* nothossp. *rotundata* (Link) Janchen et Neumayer (syn. *P. uliginosa* Neumann), which is believed to have arisen from hybridization of the two former species [8,9], but a non-hybrid origin has also been suggested [10,11]. In the pure populations, *P. uncinata* may be found in the Pyrenees, western Alps, and in eastern Switzerland [12]. In its typical form, *P. rotundata* occurs on the foothills of the northern side of the Alps with the center of its distribution in the southwestern and southern parts of the Czech Republic. According to Businsky [7], all data and reports on the presence of *P. rotundata* in the Carpathian region are incorrect. It is the hybrid swarms of *P. sylvestris* × *mugo* s. str. occurring in the region that have been mistakenly taken for *P. rotundata*. Their occurrence has been reported in Czech Republic [7], Bulgaria [13,14], and Poland [15,16]. In Slovakia, the hybrid combination *P. sylvestris* × *P. mugo* s. str. may be found primarily in the northern part of the country, where the areas of the parental species overlap. The hybrids are considered here to be the products of the most recent hybridization events taking place on individual sites with a varying intensity and extent. Their habitats involve the peat bogs near Zuberec (Medzi bormi), Sucha Hora (Rudne), and Oravska Polhora (Tisovnica), as well as a calcareous ravine near Terchova (Obsivanka). Of these, Tisovnica and Zuberec are supposed to represent the hybrid zones with advanced introgression, whereas the remaining localities should contain only isolated hybrid trees [7].

To provide evidence of *P. sylvestris* × *mugo* hybridization at the localities, several studies on paternally inherited chloroplast DNA markers were conducted in the past, revealing significant proportions of hybrid embryos, especially in Zuberec (41.1–58.7%) and Obsivanka (5–17.5%) [17,18]. These markers could also be used to test for hybridity of individual trees. However, with species-specific mitochondrial DNA markers being absent, there is no way to identify maternal parent taxonomically. Regardless, the suitability of organelle DNA markers for identification of interspecific heterozygotes has been challenged by the recent discovery of possible maternal inheritance of cpDNA in spontaneous hybridization [19,20]. For these reasons, the genetic status of the swarms has not been settled yet, with different opinions relative to their hybrid origin. Originally, based on morphological evaluation of the stand in Zuberec, Musil [21] postulated its hybrid nature. However, the needle anatomy data [22] have supported this conclusion only partially. The same is true of the isoenzyme study on the putative hybrid swarms in Tisovnica and Zuberec, which has not revealed the existence of a correlation between the isoenzyme-derived genotypes and the morphology of individual trees [23]. Owing to the limited number of isoenzyme systems used in the study, these results should be taken as preliminary. Still, another study on isoenzyme polymorphism, which involved the four putative hybrid swarms in Slovakia and 12 isoenzyme loci, revealed that the population in Sucha Hora represents a mixed stand, consisting of pure-species individuals of *P. mugo* and *P. sylvestris,* whereas the putative hybrid swarms in Zuberec, Tisovnica, and Obsivanka are supposed to be of hybrid origin [24].

The present study aimed to provide genomic estimates of individual admixture proportions and diversity of the swarms based on nuclear DNA as additional and clear evidence for their hybrid nature. As a multilocus approach, we have chosen inter-primer binding site (iPBS) amplification with a single PBS primer, a universal PCR-based method amplifying genomic DNA between long terminal repeat (LTR) retrotransposons in an inverted orientation [25]. According to these authors, the iPBS amplification has proved to be a powerful DNA fingerprinting technology without the need for prior sequence knowledge, due to conserved regions of the PBS domains of LTR retrotransposons. Its effectiveness is comparable to inter-retrotransposon amplification polymorphism (IRAP),

retrotransposon-microsatellite amplification polymorphism (REMAP), or sequence-specific amplified polymorphism (SSAP), but the method also reveals polymorphism in both *Gypsy* and *Copia* superfamilies of LTR retrotransposons along with non-autonomous LARD and TRIM elements.

## 2. Materials and Methods

### 2.1. Sampled Material and DNA Isolation

Genetic variations of 13 populations, including Scots pine (*Pinus sylvestris* L.) and dwarf mountain pine (*P. mugo* Turra s. str.), originating from their natural habitats, along with a group of their putative hybrid swarms occurring in the four contact zones of northern Slovakia, were analyzed. The list of populations and their locations are given in Table 1. Young needles collected from individual trees during May–August 2017–2018 served as a material for DNA extraction. The needle harvest was done randomly under consideration of a 15 m distance between trees to minimize burdening of samples by clonality or inbred strains. Collected needles were placed at −81 °C and after short-term storage were used in DNA isolation. Total DNA was isolated following the CTAB protocol [26]. DNA integrity was checked on 1% agarose gel with ethidium bromide (Figure S1, supplementary material), and concentration was assessed with a NanoDrop spectrophotometer (BioSpec-nano, Shimadzu).

**Table 1.** List of populations used in study.

| Locality | Abbr. | Area | Region | GPS Coordinates | Altitude | Subsoil |
|---|---|---|---|---|---|---|
| *P. sylvestris* | | | | | | |
| Hrustin | S/Hr | - | Orava | 49°20′ N; 19°19′ E | 830 m | mineral |
| Cierny Vah | S/CV | - | Nizke Tatry | 49°00′ N; 19°56′ E | 790 m | mineral |
| Oravsky Biely Potok | S/OP | - | Orava | 49°17′ N; 19°32′ E | 735 m | mineral |
| Strba | S/St | - | Vysoke Tatry | 49°07′ N; 20°03′ E | 1410 m | mineral |
| Hybrid swarms | | | | | | |
| Zuberec (Medzi bormi) | H/Zu | 6 ha | Orava | 49°16′ N; 19°37′ E | 817 m | peatbog |
| Sucha Hora (Rudne) | H/SH | 2 ha | Orava | 49°23′ N; 19°47′ E | 750 m | peatbog |
| Terchova (Obsivanka) | H/Ob | 21 ha | Mala Fatra | 49°14′ N; 19°01′ E | 840 m | mineral |
| Oravska Polhora (Tisovnica) | H/Ti | 15 ha | Orava | 49°33′ N; 19°23′ E | 745 m | peatbog |
| *P. mugo* | | | | | | |
| Rohace | M/Ro | - | Zapadne Tatry | 49°12′ N; 19°45′ E | 1465 m | mineral |
| Suchy | M/Su | - | Mala Fatra | 49°10′ N; 18°57′ E | 1412 m | mineral |
| Vratna dolina | M/VD | - | Mala Fatra | 49°11′ N; 19°02′ E | 1280 m | mineral |
| Skalnate Pleso | M/SP | - | Vysoke Tatry | 49°11′ N; 20°14′ E | 1734 m | mineral |
| Jasna | M/Ja | - | Nizke Tatry | 48°57′ N; 19°34′ E | 1546 m | mineral |

### 2.2. PCR Amplification

In total, 10 iPBS primers were used (Table 2). The primers were chosen according to PCR efficiency evaluated by the authors [25]. Amplification was performed in a 25 μL reaction mixture containing ~600 ng of DNA for 12 nt primers and/or ~300 ng for 18 nt primers, 1× B1 buffer, 2.5 mM MgCl2, 1 μM primer, 0.4 mM dNTPs, and 1.5 U of HOT FIREPol® DNA Polymerase (Solis BioDyne) for 12 nt primers and/or 0.5 U for 18 nt primers. The PCR program was initiated by a polymerase activation step at 95 °C (15 min), followed by 40 cycles at 95 °C (15 s)/51–63.3 °C (60 s)/72 °C (60 s), and a final extension at 72 °C for 5 min. All PCR reactions were carried out using the same source of PCR-grade water (SolisBiodyne) in the TProfessional Gradient Thermocycler (Biometra).

**Table 2.** Details of the primers used in study.

| Primer | Length [nt] | Sequence | $T_M$ [°C] | CG [%] | $T_A$ [°C] |
|---|---|---|---|---|---|
| 2077 | 12 | 5′-CTCACGATGCCA-3′ | 46.1 | 58.3 | 55.1 |
| 2080 | 12 | 5′-CAGACGGCGCCA-3′ | 54.6 | 75.0 | 63.3 |
| 2083 | 12 | 5′-CTTCTAGCGCCA-3′ | 45.7 | 58.3 | 54.6 |
| 2374 | 12 | 5′-CCCAGCAAACCA-3′ | 47.1 | 58.3 | 53.5 |
| 2378 | 12 | 5′-GGTCCTCATCCA-3′ | 44.2 | 58.3 | 53.0 |
| 2224 | 18 | 5′-ATCCTGGCAATGGAACCA-3′ | 56.6 | 50.0 | 55.4 |
| 2237 | 18 | 5′-CCCCTACCTGGCGTGCCA-3′ | 65.0 | 72.2 | 55.0 |
| 2239 | 18 | 5′-ACCTAGGCTCGGATGCCA-3′ | 60.4 | 61.1 | 55.0 |
| 2242 | 18 | 5′-GCCCCATGGTGGGCGCCA-3′ | 69.2 | 77.8 | 57.0 |
| 2373 | 18 | 5′-GAACTTGCTCCGATGCCA-3′ | 57.9 | 55.6 | 51.0 |

*2.3. DNA Fragment Analysis and Data Collection*

The PCR products were separated in 1.7% agarose gels with ethidium bromide and 1× TBE buffer. The gels were run at 80 V for 7 h (3.2 V/cm) and photographed by BioDoc-It (UVP) (Figures S2–S6, Supplementary Material). Bands were estimated using 100 bp DNA Ladder (Solis Biodyne) and scored for their presence (1) or absence (0). Only the repeatable marker data based on polymorphic bands were taken into account during statistical processing. Unreliable bands with a wide range of band intensities (e.g., PCR artifacts) were identified by repeatability tests and excluded from the study (Figure 1). In addition, poorly separated and monomorphic bands of the total frequency above 1–(3/N), where N is total number of individuals, were excluded from the estimation due to the potential for upward bias of diversity estimates [27]. The whole process of gel electrophoresis interpretation was done two times by a single person to reduce subjective bias.

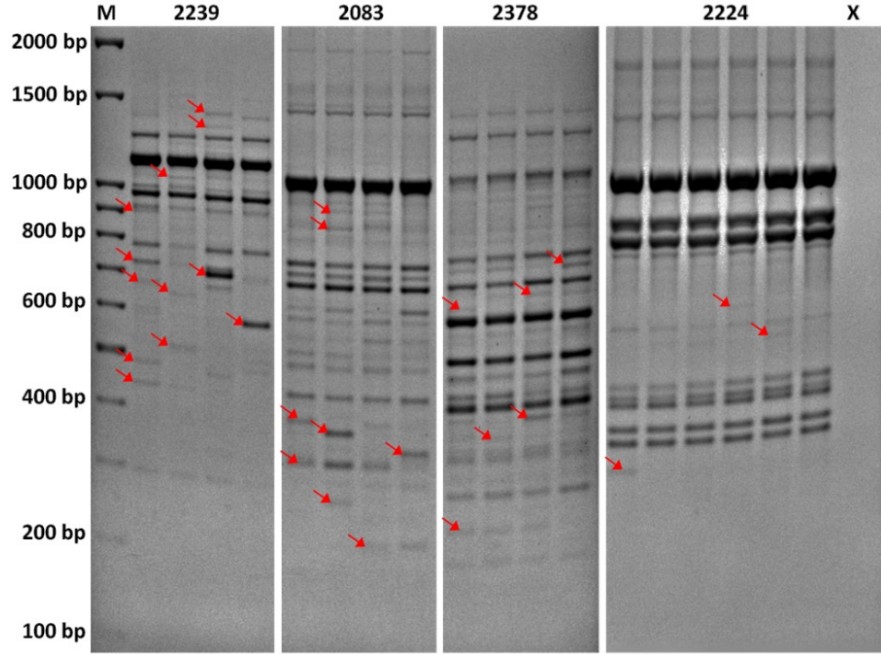

**Figure 1.** Repeatability of DNA profiles by iPBS primers 2239, 2083, 2378, and 2224. The lanes represent DNA fingerprints of a single individual. To identify unreliable regions of DNA profiles, the testing for differences in banding patterns was carried out by repeated PCR with a single DNA sample for each of the primers, a few times after PCR optimization. This information was used in gel interpretation to better score for the presence and absence of individual bands in different regions. The bands considered unreliable with low repeatability are indicated by the arrows.

*2.4. Statistics*

The iPBS marker data were statistically analyzed taking mean values of population genetic parameters across all loci examined. Firstly, genetic variation within populations was estimated using PopGene 1.32. Allelic variation was quantified as observed number of alleles ($n_A$), Nei's [28] gene diversity ($H_E$), and Kimura and Crow's [29] effective number of alleles or allelic diversity ($n_E$). The significance of differences among population groups (*P. sylvestris*, *P. mugo*, putative hybrid swarms) in their within-population variation was tested by two-way nested analysis of variance (ANOVA) and Tukey's multiple comparison test in R 4.1.1.

To describe population genetic structure in the contact zones, the multilocus hybrid index (*h*) was estimated for individual trees based on reference *P. sylvestris* and *P. mugo* allele frequencies, using the maximum likelihood method in FAMD 1.31 software. The frequency distribution of these data was tested for number of modes by six different tests: Silverman's (SI), Fisher and Maroon's (FM), Ameijeiras-Alonso's et al. (ACR), Hall and York's (HY), Hartigan and Hartigan's (HH), and Cheng and Hall's (CH) (the last three test unimodality only, see [30]). The modes were localized under unimodal assumption in order to find the most frequent form within the individual hybrid zones. Parental, introgressant, and true hybrid forms (intermediates) were defined by an interval width of 0.2 each.

Finally, analysis of molecular variance (AMOVA) conducted by FAMD was used to separate the total genetic variance into within and among population variances. The intraspecific differentiation was quantified as Nei's [28] fixation index ($G_{ST}$) by PopGene and differentiation of the putative hybrid swarms from *P. sylvestris* or *P. mugo* as Gregorius and Roberds' [31] index ($D_{K(S/M)}$), calculated in Excel. Tree diagrams were constructed based on a distance matrix of Nei's [32] unbiased measure of genetic distance (*D*), as implemented in PopGene. Cluster analysis was performed using unweighted pair group method with arithmetic means (UPGMA) and neighbor joining (NJ) was performed using Phylip 3.695. Bootstrap support of individual clades was calculated from 1000 random resamplings across the loci. The trees were generated by TreeViewX 0.5.0. Additionally, in reference populations, a locus-by-locus comparison was carried out in terms of interspecific $D_K$ and intraspecific $G_{ST}$ calculated in Excel and PopGene, respectively. In this way, the individual marker loci were tested for their taxonomic specificity.

## 3. Results

### *3.1. Population Genetic Variation*

A total of 490 samples and 10 iPBS primers yielded 212 preliminary bands, of which 132 bands were evaluated either as unreliable or monomorphic and excluded from the study. The remaining 80 bands were polymorphic with a maximum frequency of 0.942 in the entire sample. The bands or putative loci varied between 290–1250 bp in length, and were generated in different amounts by individual primers (2077–0, 2080–10, 2083–9, 2374–5, 2378–7, 2224–17, 2237–8, 2239–3, 2242–10, 2373–11).

The population genetic summary statistics are given in Table 3. It follows from presented data that mean allelic variation across loci was similar in individual populations of a given taxon. The lowest variation was found in *P. sylvestris* populations, as evidenced by the weighted mean of $n_A = 1.89$, $H_E = 0.27$, and $n_E = 1.44$. The corresponding characteristics in the *P. mugo* populations were slightly higher, averaging $n_A = 1.91$, $H_E = 0.28$, and $n_E = 1.48$. The putative hybrid populations deviated from the reference populations mentioned above, with the highest values of their allelic variation characteristics reaching mean $n_A = 1.97$, $H_E = 0.32$, and $n_E = 1.54$. No significant differences were revealed between investigated populations using nested ANOVA ($p = 0.040$ for $n_A$, $p = 0.894$ for $H_E$, and $p = 0.956$ for $n_E$), but the three groups of populations differed significantly in their within-population variation ($p = 0.009$ for $n_A$, $p < 0.001$ for $H_E$, and $p = 0.010$ for $n_E$). In addition, the Tukey's multiple comparison test confirmed that putative hybrid swarm populations were statistically more variable than those of *P. sylvestris* ($p < 0.001$ for $n_A$, $H_E$ and $n_E$) and *P. mugo* ($p = 0.012$ for $n_A$, $p = 0.018$ for $H_E$, and $p = 0.032$ for $n_E$). On the contrary, the two

groups of *P. sylvestris* and *P. mugo* reference populations did not differ significantly from each other, as evidenced by the probability values $p = 0.491$ for $n_A$, $p = 0.317$ for $H_E$, and $p = 0.267$ for $n_E$. Among the populations studied so far, the putative hybrid swarms Tisovnica and Sucha Hora were most variable ($n_A = 1.99$, $H_E = 0.33$, $n_E = 1.57$ and $n_A = 1.98$, $H_E = 0.32$, $n_E = 1.55$, respectively).

**Table 3.** Population genetic summary statistics for *P. sylvestris*, *P. mugo*, and their putative hybrid swarms in Slovakia (pop. abbreviations in Table 1).

| Pop. | Within-Population Variation | | | | Among-Population Variation | | |
|---|---|---|---|---|---|---|---|
| | $N \pm$ SD | $n_A \pm$ SD | $H_E \pm$ SD | $n_E \pm$ SD | $G_{ST}$ | $D_{K(S)} \pm$ SD | $D_{K(M)} \pm$ SD |
| S/Hr | 31 ±0 | 1.86 ± 0.35 | 0.25 ± 0.18 | 1.41 ± 0.35 | | - | - |
| S/CV | 34 ± 0 | 1.91 ± 0.28 | 0.27 ± 0.17 | 1.44 ± 0.33 | 0.0377 | - | - |
| S/OP | 30 ± 0 | 1.86 ± 0.35 | 0.27 ± 0.17 | 1.44 ± 0.34 | | - | - |
| S/St | 35 ± 0 | 1.90 ± 0.30 | 0.28 ± 0.17 | 1.46 ± 0.34 | | - | - |
| WM | | 1.89 ± 0.03 | 0.27 ± 0.01 | 1.44 ± 0.02 | | | |
| H/Zu | 39 ± 2 | 1.96 ± 0.19 | 0.31 ± 0.16 | 1.52 ± 0.33 | | 0.10 ± 0.10 | 0.23 ± 0.19 |
| H/SH | 44 ± 0 | 1.98 ± 0.16 | 0.32 ± 0.16 | 1.55 ± 0.34 | 0.0799 | 0.19 ± 0.15 | 0.13 ± 0.11 |
| H/Ob | 42 ± 0 | 1.94 ± 0.24 | 0.31 ± 0.17 | 1.53 ± 0.35 | | 0.21 ± 0.16 | 0.12 ± 0.12 |
| H/Ti | 48 ± 0 | 1.99 ± 0.11 | 0.33 ± 0.15 | 1.57 ± 0.33 | | 0.23 ± 0.18 | 0.12 ± 0.11 |
| WM | | 1.97 ± 0.02 | 0.32 ± 0.01 | 1.54 ± 0.02 | | | |
| M/Ro | 33 ± 0 | 1.83 ± 0.38 | 0.27 ± 0.19 | 1.46 ± 0.38 | | - | - |
| M/Su | 34 ± 0 | 1.86 ± 0.35 | 0.27 ± 0.18 | 1.45 ± 0.34 | | - | - |
| M/VD | 42 ± 1 | 1.95 ± 0.22 | 0.30 ± 0.17 | 1.51 ± 0.34 | 0.0383 | - | - |
| M/SP | 35 ± 0 | 1.95 ± 0.22 | 0.29 ± 0.17 | 1.47 ± 0.34 | | - | - |
| M/Ja | 40 ± 0 | 1.95 ± 0.22 | 0.30 ± 0.17 | 1.49 ± 0.33 | | - | - |
| WM | | 1.91 ± 0.06 | 0.28 ± 0.01 | 1.48 ± 0.02 | | | |

The mean values across 80 polymorphic iPBS loci are shown; *N*—sample size, $n_A$—observed number of alleles, $H_E$—gene diversity, $n_E$—effective number of alleles; $G_{ST}$—fixation index; $D_{K(S/M)}$—genetic differentiation from *P. sylvestris* or *P. mugo* population group; SD—standard deviation; WM—weighted mean by sample size.

### 3.2. Population Genetic Structure

Two distinct patterns emerged from the 80-loci hybrid index (*h*) data when considering range, distribution, and mode position in individual hybrid zone populations (Figure 2). The putative hybrid swarm Zuberec (H/Zu) was the most conspicuous in this respect, with *h* values between 0.43 and 1 (zero indicates pure *P. mugo*, one stands for pure *P. sylvestris*). The corresponding values in the remaining three swarms ranged between 0–0.57 (H/Ti), 0–0.73 (H/SH), and 0–0.69 (H/Ob). Statistical tests SI, FM, ACR, HY, CH, and HH supported unimodal distribution in the three datasets, especially in H/SH and H/Ob (Figure 2). Their modes were very similar, deviating by −0.182 (H/Ti), −0.183 (H/SH), and −0.112 (H/Ob) from the value 0.5, being closer to the midpoint than the mode of H/Zu. The unimodality of the latter was unclear ($p = 0.042$ of ACR, $p = 0.026$ of CH) but still significant for the remaining four tests, with a peak deviated by +0.265.

When defining individual forms by an interval width of 0.2 (Figure 3), the most frequent forms in the H/Zu swarm were trees of pure *P. sylvestris* (46.3%) and *P. sylvestris*-like introgressants (39.0%), which predominated considerably over true hybrids (14.6%). A similar situation was found in the H/Ti population, with *P. mugo* as the only parental form present (37.5% of pure *P. mugo*, 50% of *P. mugo*-like introgressants, 12.5% of intermediates). The remaining *P.mugo*-like swarms were characterized by distinct genetic structures. In particular, the parental form here was lower (H/SH-13.6%, H/Ob-26.2%) and intermediates were higher (H/SH-29.5%, H/Ob-28.6%), relative to these frequencies in H/Zu and H/Ti. Of the four hybrid zone populations, the lowest difference between the frequencies of introgressant and intermediate forms was found in H/Ob (42.9% vs. 28.6%, respectively).

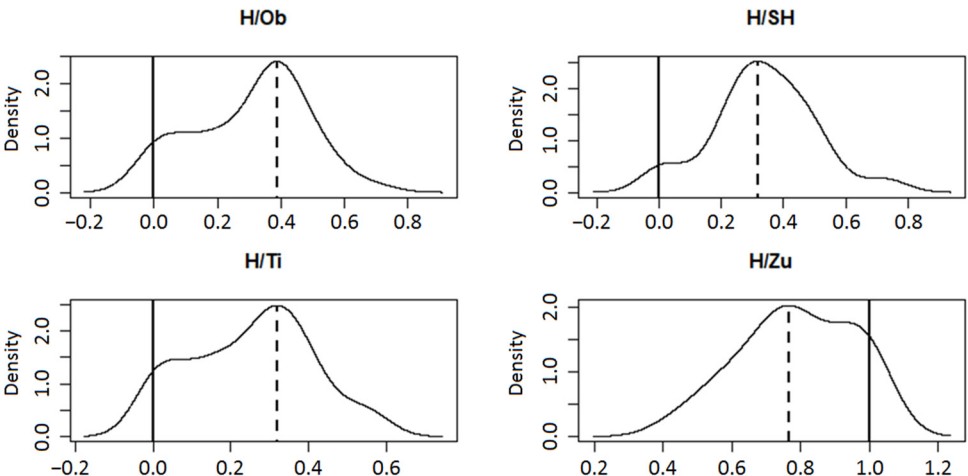

**Figure 2.** Hybrid index calculated for 80 iPBS loci in putative hybrid swarms Obsivanka (H/Ob), Sucha Hora (H/SH), Tisovnica (H/Ti), and Zuberec (H/Zu) (zero indicates pure *P. mugo*, one stands for pure *P. sylvestris*). Density function with mode localization for a unimodal solution is shown. The dashed verticals represent the mode and the solid verticals represent the second possible mode.

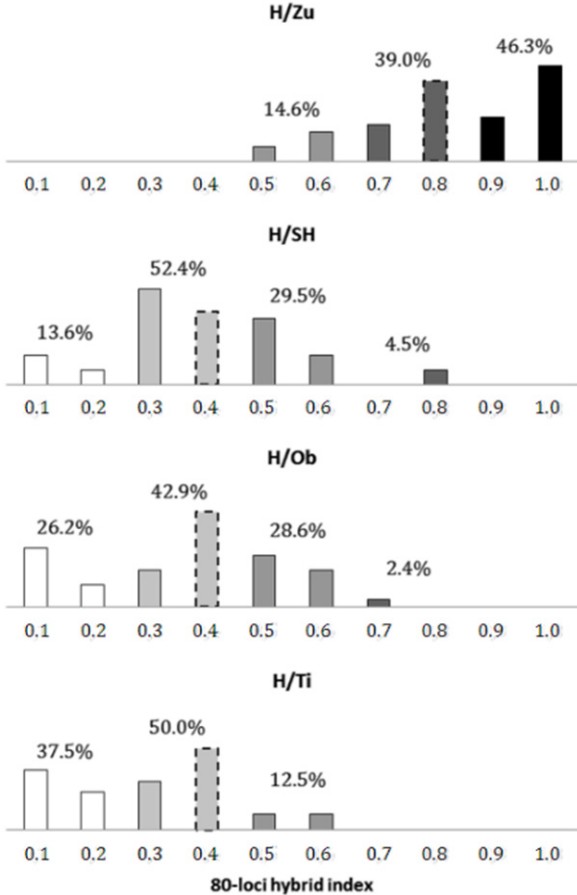

**Figure 3.** Frequency distribution of individual forms within putative hybrid swarms *P. sylvestris* x *mugo* in Slovakia. Genotypes considered as parental taxa are shown in black (*P. sylvestris*) and white (*P. mugo*), while introgressant and intermediate forms are shown in gray shades. The dashed bars represent the mode localizations as determined above.

Hybrid index data referring to all the populations including reference species are presented in the (supplementary material Figure S7).

### 3.3. Genetic Differentiation and Phylogeny

Across all iPBS loci and individuals, as much as 27.35% of the total variation is explained by an among-population component, as shown by AMOVA. However, the mean fixation indices across loci were only at $G_{ST} < 0.05$ in both *P. sylvestris* and *P. mugo* groups, i.e., 0.0377 in the former and 0.0383 in the latter (Table 3). The mean $G_{ST}$ value was two times higher in the group of putative hybrid swarms (0.0799), which was caused almost entirely by the H/Zu population. As described above, the genetic structure of this population deviated significantly from those of H/Ti, H/Ob, and H/SH. Compared to putative parental species, the differentiation of H/Zu was estimated to be two times lower in relation to *P. sylvestris* ($D_{K(S)} = 0.10$) than to *P. mugo* ($D_{K(M)} = 0.23$), while the opposite was true for H/Ti ($D_{K(S/M)} = 0.23/12$), H/Ob ($D_{K(S/M)} = 0.21/12$), and H/SH ($D_{K(S/M)} = 0.19/13$). The difference among the latter three (i.e., *P. mugo*-like swarms) was found in respect to differentiation from *P. sylvestris* only, which was most evident at the population level. As shown in Table 4, the genetic distance between them and *P. sylvestris* populations varied within the range of 0.122–0.139 in H/Ti, 0.093–0.112 in H/Ob, and 0.082–0.096 in H/SH, but the estimates were remarkably similar with respect to *P. mugo* (0.033–0.052, 0.034–0.052, and 0.035–0.052, respectively). Likewise, the genetic distance between H/Zu and *P. mugo* varied more (0.125–0.162) than the genetic distance between H/Zu and *P. sylvestris* (0.027–0.033), the species to which the population belongs.

**Table 4.** Genetic distance matrix calculated as Nei's [32] unbiased measure of genetic distance.

| | S/Hr | S/CV | S/OP | S/St | H/Zu | H/SH | H/Ob | H/Ti | M/Ro | M/Su | M/VD | M/SP | M/Ja |
|---|---|---|---|---|---|---|---|---|---|---|---|---|---|
| **S/Hr** | | | | | | | | | | | | | |
| **S/CV** | 0.012 | | | | | | | | | | | | |
| **S/OP** | 0.014 | 0.012 | | | | | | | | | | | |
| **S/St** | 0.019 | 0.007 | 0.015 | | | | | | | | | | |
| **H/Zu** | 0.027 | 0.031 | 0.033 | 0.032 | | | | | | | | | |
| **H/SH** | 0.096 | 0.082 | 0.091 | 0.091 | 0.057 | | | | | | | | |
| **H/Ob** | 0.112 | 0.093 | 0.105 | 0.109 | 0.073 | 0.024 | | | | | | | |
| **H/Ti** | 0.139 | 0.122 | 0.131 | 0.128 | 0.080 | 0.028 | 0.038 | | | | | | |
| **M/Ro** | 0.246 | 0.218 | 0.235 | 0.231 | 0.162 | 0.052 | 0.052 | 0.052 | | | | | |
| **M/Su** | 0.239 | 0.201 | 0.227 | 0.217 | 0.159 | 0.050 | 0.050 | 0.047 | 0.015 | | | | |
| **M/VD** | 0.198 | 0.173 | 0.194 | 0.185 | 0.125 | 0.035 | 0.040 | 0.034 | 0.020 | 0.015 | | | |
| **M/SP** | 0.201 | 0.169 | 0.198 | 0.180 | 0.128 | 0.036 | 0.034 | 0.033 | 0.021 | 0.014 | 0.010 | | |
| **M/Ja** | 0.218 | 0.191 | 0.215 | 0.202 | 0.142 | 0.041 | 0.044 | 0.036 | 0.020 | 0.011 | 0.009 | 0.009 | |

Based on UPGMA clustering, the two distinct groups were identified among the studied populations, separating the *P. sylvestris* outgroup and the H/Zu sister group from the remaining ingroup and *P. mugo* outgroup (Figure 4). The only sister group to the rest of the species (i.e., the first outgroup) was found in the *P. mugo* outgroup (M/Ro).

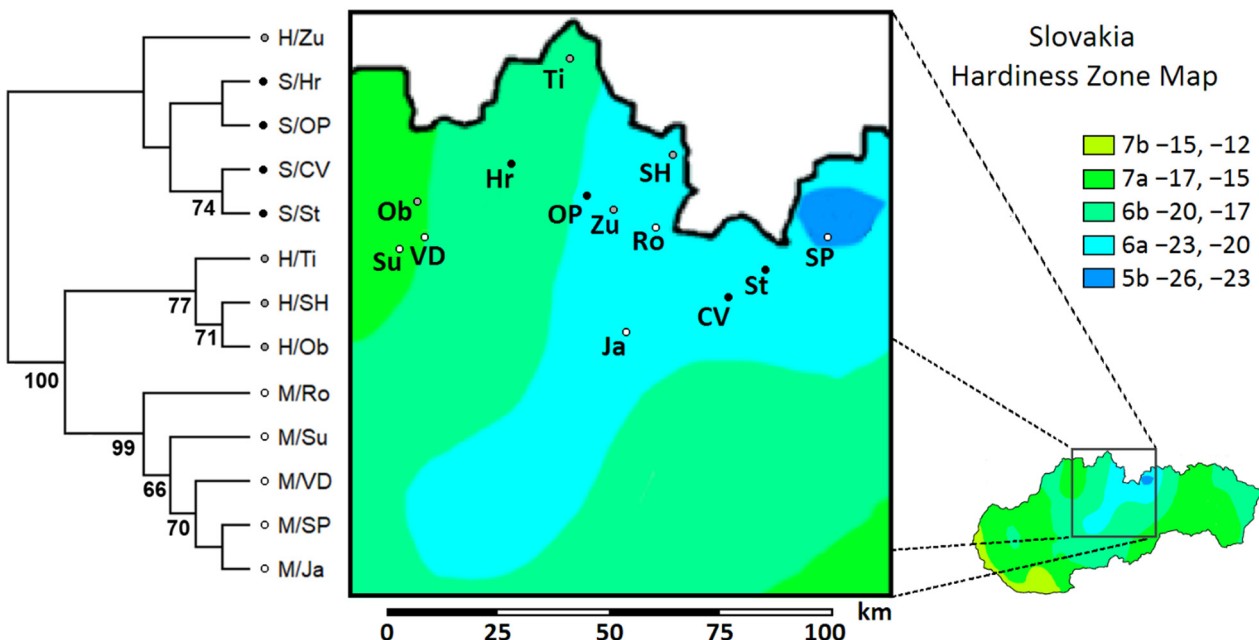

**Figure 4.** UPGMA distance phenogram illustrating genetic similarity among Slovak populations of *P. sylvestris*, *P. mugo*, and their putative hybrids swarms in 80 polymorphic iPBS loci. Bootstrap support >50% is shown. (Pop. abbreviations in Table 1; map available online: https://commons. wikimedia.org/wiki/File:Slovakia_Hardiness_Zones.png#filelinks (accessed on 17 October 2021), License available online: https://creativecommons.org/licenses/by-sa/4.0 (accessed on 17 October 2021).

By contrast, the neighbor-joining phylogram, also illustrating genetic change within branches, showed that the most diverged *P. mugo* population was M/Ro, but the closest lineage to the *P. mugo*-like ingroup (H/Ti, H/Ob, H/SH) was M/SP (Figure 5). Additionally, the S/CV population was recognized as the first *P. sylvestris* outgroup and the closest relative to H/Zu. Out of all putative hybrid swarms, H/Ti represented the longest branch.

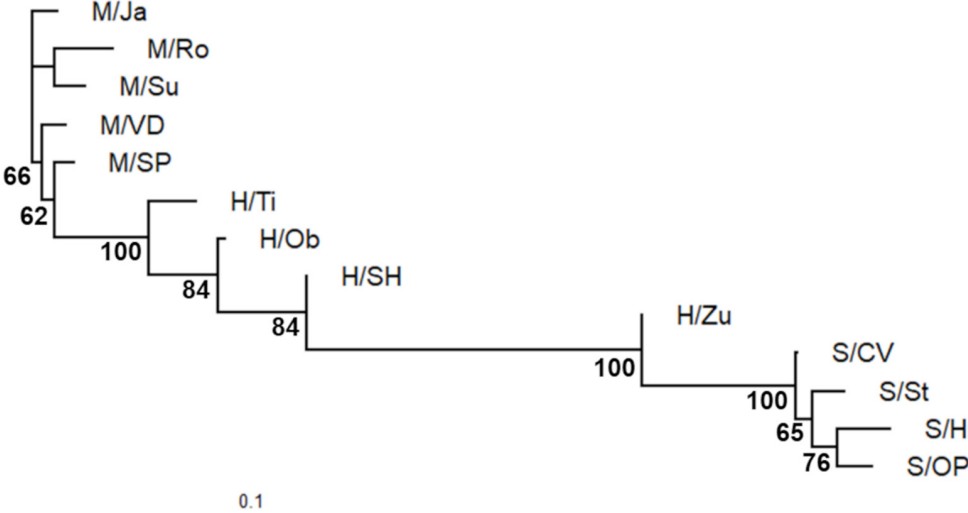

**Figure 5.** Neighbor-joining phylogram showing phylogenetic relationships among studied populations (pop. abbreviations in Table 1). The tree relies on the same distance matrix as in UPGMA clustering. Bootstrap support >50% is shown.

### 3.4. Inter- and Intraspecific Differentiation by Individual iPBS Loci

A great total differentiation was observed among all reference populations when calculating $G_{ST}$ as average per locus ($G_{ST(SM)}$ = 0.212) (Table 5). Preferentially, this variation was caused by the differences between *P. sylvestris* and *P. mugo*, both of which had $G_{ST}$ values below 0.05 on average (Table 3). Not only were the species separated by each of the nine primers used in UPGMA, but the locus-by-locus evaluation also showed that out of 80 loci, only 13 (16.25%) were a little differentiated ($G_{ST(SM)}$ < 0.05), while 35 (43.75%), 14 (17.50%), and 18 (22.50%) loci exhibited moderate ($G_{ST(SM)}$ = 0.05–0.15), great ($G_{ST(SM)}$ = 0.15–0.25), and very great differentiation ($G_{ST(SM)}$ > 0.25), respectively (Table 5). In the two most differentiated loci, 2374–540 bp and 2373–1040 bp, the $D_K$ values between species reached ~0.9, indicating the potential for species-diagnostic nDNA markers. In the first locus, i.e., 2374–540 bp, the *P. sylvestris* populations were fixed for the allele 1, whereas in *P. mugo,* there was a weighted mean frequency of 0.04 among the populations and $G_{ST}$ of 0.0047. On the other hand, in the second most differentiated locus, i.e., 2373–1040 bp, the *P. mugo* populations were fixed and the *P. sylvestris* populations had a weighted frequency of 0.13, with $G_{ST}$ of 0.0011 only.

**Table 5.** Summary of iPBS locus-by-locus evaluation of genetic differentiation within and between *P. sylvestris* and *P. mugo* in Slovakia.

| Number/ Name of Loci | $p_S \pm$ SD | $p_M \pm$ SD | $D_K$ | $G_{ST(S)}$ | $G_{ST(M)}$ | $G_{ST(SM)}$ |
|---|---|---|---|---|---|---|
| 13 (16.25%) | | | | | | <0.05 |
| 35 (43.75%) | | | | | | 0.05–0.15 |
| 14 (17.50%) | | | | | | 0.15–0.25 |
| 18 (22.50%) | | | | | | >0.25 |
| Total (100%) | | | | | | 0.2120 |
| 2373–1040 bp | 0.13 ± 0.01 | 1.00 ± 0.00 | 0.87 | 0.0011 | - | 0.7853 |
| 2374–540 bp | 1.00 ± 0.00 | 0.04 ± 0.01 | 0.96 | - | 0.0047 | 0.9206 |

$p_S$ and $p_M$—weighted means of allele frequencies (allele 1) in the four *P. sylvestris* and the five *P. mugo* populations; $D_K$—interspecific differentiation; $G_{ST(S)}$ and $G_{ST(M)}$—intraspecific differentiation; $G_{ST(SM)}$—total differentiation; SD—standard deviation.

## 4. Discussion

Despite clear morphological and ecological differentiation between *P. sylvestris*, *P. mugo*, *P. uncinata,* and *P. uliginosa*, the analysis of nuclear genes showed that these species share a similar genetic background [33]. Using transcriptome sequencing [34], the highest pairwise nucleotide divergence was found between *P. mugo* and *P. sylvestris,* along with a closer genetic relationship between *P. mugo* and *P. uliginosa* as compared to *P. sylvestris*. No significant genetic differentiation was revealed between *P. mugo* and *P. uncinata* vs. *P. uliginosa.* Moreover, the three species of the *P. mugo* complex were shown to share the same haplotypes of chloroplast and mitochondrial DNAs [35]. The extensive DNA barcoding approach, involving eight chloroplast DNA regions, was shown to be ineffective in distinguishing closely related pines from the *P. mugo* complex. The discriminating power of barcoding regions equaled zero [36]. Owing to the reduced potential of the chloroplast and mitochondrial DNA in developing the species-specific markers, the iPBS amplification was applied in the present study oriented towards the analysis of genetic structure of *P. sylvestris* and *P. mugo* populations and their putative hybrid swarms. This method is supposed to provide major insight into neutral introgression and phylogeny, mainly because the estimates are based on many neutral loci that likely represent different genomic regions. It is believed that genetic distance estimates derived in this way should be more accurate and less contaminated by convergence or parallel evolution, reflecting phylogenetic rather than phenetic relationships. Apart from this ability to reduce bias by chance (when allelic differences among populations do not correspond to their phylogenetic branching order), standard PCR-based multilocus techniques are generally supposed to be

more suitable in assessing phylogeny than single-locus approaches due to horizontal gene transfer and unequal vertical transfer of gene polymorphism [37].

### 4.1. Within-Population Variation

Our data, derived from 80 iPBS loci, indicate that the putative hybrid swarms are more variable than natural stands of pure *P. sylvestris* and *P. mugo* from adjoining mountain ranges, including Mala Fatra, Tatra Mts., and the Orava region. The same was reported for the progeny of natural hybrids between *P. sylvestris* and *P. mugo* from the peat-bog "Bor na Czerwonem" in Poland, where the hybrid population exhibited a much higher level of variability than pure populations of the parental species [38]. As far as allelic richness and diversity are concerned, the most noticeable differences were not found on Zuberec locality, the hybrid nature of which is the most apparent according to the habitus of trees, cone morphology, and needle anatomy [22,39,40], but, surprisingly, in Tisovnica and Sucha Hora, which exhibited the highest allelic variation. However, the last mentioned population seems to be controversial in this context, as evidenced by the isoenzyme data, which deny its hybridity [24].

### 4.2. Status of Hybrid Zone Populations

There exist several reasons to believe that a simple hybrid index model is of better use than the extensive alternative Bayesian approach implemented in Structure [41]. One of the reason is that admixture model underlying Structure cannot utilize information from dominant markers, which may be informative in later generation of hybrids. Therefore, based on the presence of intermediate admixture proportions calculated via a 80-loci hybrid index, we are on firm ground in postulating that each of the study swarms is of hybrid origin. However, the hybrid swarm status defined strictly as a unimodal hybrid zone with a peak representing intermediate hybrids [42] cannot be ascribed to these populations. Instead, they are more or less introgressed to one or other parental species, each of them containing introgressive hybrids as the most frequent form. When estimating magnitude of introgression based on deviation of the peak from the midpoint, the highest amount of backcrossing seems to have occurred within the *P. sylvestris*-like population in Zuberec. An opposite pattern of substantial introgression towards *P. mugo* is evident in Tisovnica and Sucha Hora, the latter being less distinct in this respect when taking into account the difference between the frequencies of introgressive and true intermediate hybrids. The least introgressed appears to be the population of Obsivanka, as evidenced by the peak at the boundary between *P. mugo*-like introgressants and intermediates. A relatively high frequency of intermediates in Sucha Hora and Obsivanka may also indicate a transitional state between the early generation hybrid swarm and the introgressed population of *P. mugo*, irrespective of the patterns of morphological variation within these populations. Several studies on cpDNA inheritance during the process of hybridization between *P. sylvestris* and *P. mugo* showed that gene flow by pollen is more frequent from *P. mugo* to *P. sylvestris* than in opposite direction [43,44], which may be the cause for a higher proportion of *P. mugo* genes in the three putative hybrid swarms mentioned above.

### 4.3. Genetic Differentiation and Phylogeny

An earlier study on genetic differentiation of *P. sylvestris* and *P. mugo* hybrid swarms in Slovakia has confirmed their hybridity [24]. The only exception was Sucha Hora, in which no difference from *P. mugo* was revealed based on semidiagnostic allele frequencies within 12 isoenzyme loci. The population was, therefore, looked upon as a mixed stand of pure-species individuals. However, the estimate of genetic distance between Sucha Hora and *P. mugo* populations (0.021 on average) was not different from that of Tisovnica (0.023) or Obsivanka (0.023). More importantly, its differentiation from *P. sylvestris* was even lower than in the case of Tisovnica (0.039 vs. 0.053). Accordingly, based on differentiation estimates, which included both putative parental species, Sucha Hora population should be considered to be of hybrid origin more than Tisovnica. Likewise, our results referring

to 80 iPBS loci indicate that *P. mugo*-like hybrid populations Tisovnica, Obsivanka, and Sucha Hora are differentiated from *P. mugo* to an almost identical extent (~0.042), but Sucha Hora is, in fact, less differentiated from *P. sylvestris* (0.090) than both Tisovnica (0.130) and Obsivanka (0.105), therefore placing it more to the middle of NJ phylogram. However, an intermediate position in a dendrogram may also result for a mixed stand of the pure-species individuals with no hybrids present in the population. The fact that the intermediate positions of all the four populations in the phylogram are due to interspecific heterozygotes is supported by the corresponding hybrid indices. Most probably, the outlined contradictions between differentiation estimates may be ascribed to the different types of polymorphism in a genome (neutral vs. selective) and different number of loci scored. It is reasonable to assume that neutral or nearly neutral molecular markers are not likely to provide a strong evidence for hybridization between very closely related species that diverged no earlier than ~5 Ma if only a small fraction of genome is examined [33,45]. This should be particularly true of the loci underlying or linked to ecologically important traits, such as plant life form (polycormy vs. monocormy). Given that these morphological differences between *P. mugo* and *P. uliginosa* are most likely due to a very limited genomic region or are conditioned ecologically, the individuals of *P. mugo* morphology growing in a specific environment may still contain a great genomic portion of hybrid architecture [33,45–47].

**5. Conclusions**

As a continuation of the previous studies on taxonomic status of the putative hybrid swarms of *P. sylvestris* and *P. mugo* in Slovakia, which were based on growth habits of the trees, their cone shapes, needle anatomy, and isoenzyme variation, the first attempt of its kind has been made using anonymous PCR-based multilocus markers. The main outcome of the study is the confirmation of the hybrid nature of all the four swarms under comparison. This may be considered as a significant contribution to the discussion on genetic peculiarity of the trees in contact zones of the parental species in the country. However, some doubts still remain as to the degree of hybridity in individual hybrid swarms. In order to resolve this aspect of their genetic status more convincingly, additional analyses involving nuclear microsatellites and/or their sequencing are necessary.

**Supplementary Materials:** The following supporting information can be downloaded at: https://www.mdpi.com/article/10.3390/f13020205/s1, Figure S1: Quality control of DNA samples evaluated through gel electrophoresis. 1, 11, 12, 14, 15, 18, and 19—DNA samples excluded from the study; Figure S2: DNA profiles generated by iPBS primer 2080. M—size standard, lanes from the left—S/St, H/Ob, M/SP; Figure S3: DNA profiles generated by iPBS primers 2083 and 2374. M—size standard, lanes from the left—S/Hr, H/Zu, M/Ro and S/St, H/Ob, M/SP; Figure S4: DNA profiles generated by iPBS primers 2378 and 2224. M—size standard, lanes from the left—S/St, H/Ob, M/SP and S/St, H/Ob, M/SP; Figure S5: DNA profiles generated by iPBS primers 2237 and 2239. M—size standard, lanes from the left—M/Ja and S/OP, H/Ti, M/VD; Figure S6: DNA profiles generated by iPBS primers 2242 and 2373. M—size standard, lanes from the left—S/CV, H/SH, M/Su and S/St, H/Ob, M/SP; Figure S7: Boxplot showing Maximum Likelihood hybrid index of 13 pine populations in Slovakia including *P. sylvestris* (Hrustin, Cierny Vah, Oravsky Biely Potok, Strba), *P. mugo* (Rohace, Suchy, Vratna dolina, Skalnate Pleso, Jasna), and their hybrid zone populations (Zuberec, Sucha Hora, Obsivanka, Tisovnica). Hybrid index scores were calculated based on 80 iPBS marker loci, with pure *P. mugo* equalling to zero. However, considering within-species variation, we suggest defining pure parental forms as well as introgressive hybrids and intermediates by the interval of 0.2. The presence of outliers (dots) in pure populations may represent technical errors in genotyping.

**Author Contributions:** Conceptualization, M.K., M.G. and A.K.; material collection, M.G. and A.K.; DNA extraction, M.G. and A.K.; genotyping, M.K.; statistical processing, M.K. and D.G.; writing—original draft preparation, M.K.; writing—review and editing, M.K., A.K. and M.G.; project administration and supervision, A.K.; funding acquisition, A.K. All authors have read and agreed with the published version of the manuscript.

**Funding:** This research was funded by VEGA.

**Institutional Review Board Statement:** Not applicable.

**Informed Consent Statement:** Not applicable.

**Data Availability Statement:** Not applicable.

**Acknowledgments:** This study was financially supported by the VEGA Grant Agency, project no. 2/0022/20. A special thanks to rer. nat. Lubomir Rybansky, for additional statistical processing of the experimental data.

**Conflicts of Interest:** The authors declare no conflict of interest.

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
