# Peer review of "Molecular Insight into Genetic Structure and Diversity of Putative Hybrid Swarms of Pinus sylvestris × P. mugo in Slovakia"

_forests, doi:10.3390/f13020205_

Round 1

Reviewer 1 Report

The manuscript presents a well-done analysis. The authors used iPBS marker loci to analyze the population genetics and admixture proportions in 13 populations including pure /reference populations and hybrid swarms. After DNA fragment analysis, the authors identified 80 polymorphic loci, and used these 80 loci to calculate allele frequencies, population genetic structure, and genetic differentiation. In doing so, the authors infer the structure and extent of introgression in hybrid swarms. Overall, this manuscript presents a sound and solid analysis. It does not only provide insight to hybrid swarms but also a cost-effective methodology for studying conifer genetics. 

Major comments

  1. I think the concept of “iPBS” needs to be explained more in “Introduction” section. Also, in Line 89, why did you choose to use these 10 iPBS primers not others? Have you tested other primers? Selection of these 10 primers needs to be addressed clearly.
  2. For figure 2, I suggest the authors present hybrid index density graph for pure /reference populations, too, as “control”. This can be supplementary materials. In case there is distinct patterns comparing pure and hybrid populations, it would add strong evidence to support the conclusion.

Minor comments

  1. The “Introduction” section doesn’t have a clear structure. I suggest authors consider separating the “Introduction” section to three paragraphs --- “what we know”, “what we don’t know”, and “general goal and rationale of this study”.
  2. Table 1 is clear, but not straightforward. A geographical map showing the sampling locations would be much more explicit than the table.
  3. Line146: Can authors show the gel image with 80 genotyped loci, please? This can be supplementary materials, or if the size is too large, it can be stored in public repository like DRYAD.

Reviewer 2 Report

Comments are in the attached WORD file.

Round 2

Reviewer 2 Report

The manuscript has been significantly improved with all major concerns addressed. I find it suitable for publication in its present form.